# TinyGPT-V: Efficient Multimodal Large Language Model via Small Backbones

## Abstract

In recent years, multimodal large language models (MLLMs) such as GPT-4V have demonstrated remarkable advancements, excelling in a variety of vision-language tasks. Despite their prowess, the closed-source nature and computational demands of such models limit their accessibility and applicability. This study introduces TinyGPT-V, a novel open-source MLLM, designed for efficient training and inference across various vision-language tasks, including image captioning (IC) and visual question answering (VQA). Leveraging a compact yet powerful architecture, TinyGPT-V integrates the Phi-2 language model with pretrained vision encoders, utilizing a unique mapping module for visual and linguistic information fusion. With a training regimen optimized for small backbones and employing a diverse dataset amalgam, TinyGPT-V requires significantly lower computational resources—24GB for training and as little as 8GB for inference—without compromising on performance. Our experiments demonstrate that TinyGPT-V, with its language model 2.8 billion parameters, achieves comparable results in VQA and image inference tasks to its larger counterparts while being uniquely suited for deployment on resource-constrained devices through innovative quantization techniques. This work not only paves the way for more accessible and efficient MLLMs but also underscores the potential of smaller, optimized models in bridging the gap between high performance and computational efficiency in real-world applications. Additionally, this paper introduces a new approach to multimodal large language models using smaller backbones. Our code and training weights are available in the supplementary material.

---

[1]Anonymous Institution, Anonymous City, Anonymous Region, Anonymous Country. Correspondence to: Anonymous Author <anon.email@domain.com>.

Submitted to the Workshop on Advancing Neural Network Training at International Conference on Machine Learning (WANT@ICML 2024). Do not distribute.

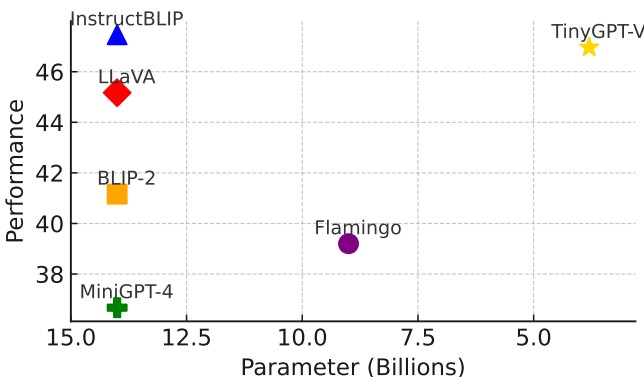

Figure 1: Comparison of TinyGPT-V with other current MLLMs models shows TinyGPT-V achieves cost-effective, efficient, and high-performing with fewer parameters.

## 1. Introduction

In recent years, the field of artificial intelligence has seen significant advancements through the development of multimodal large language models (MLLMs), such as GPT-4V, which have shown exceptional performance across a range of vision-language tasks (Yang et al., 2023). Despite GPT-4V's impressive capabilities, its closed-source nature limits its widespread application and adaptability. In contrast, the open-source landscape for MLLMs is rapidly evolving, presenting models like LLaVA and MiniGPT-4 that excel in image captioning (IC), visual question answering (VQA) often comparable GPT-4V in these areas (Dai et al., 2023; Liu et al., 2023a;b; Zhu et al., 2023). Notably, MiniGPT-v2 (Chen et al., 2023) has demonstrated superior performance in various visual grounding and question-answering tasks. However, its training code remains proprietary, which poses challenges for community-driven advancements and adaptability.

Although the impressive vision-language capabilities demonstrated by some open-source MLLMs, they frequently necessitate significant computational resources for training and inference. For example, training LLaVA-v1.5-13B (Liu et al., 2023a) required 8 × A100 GPUs, each equipped with 80GB of memory, cumulating in 25.5 hours of continuous training. As shown in Figure 2 (a), the un-

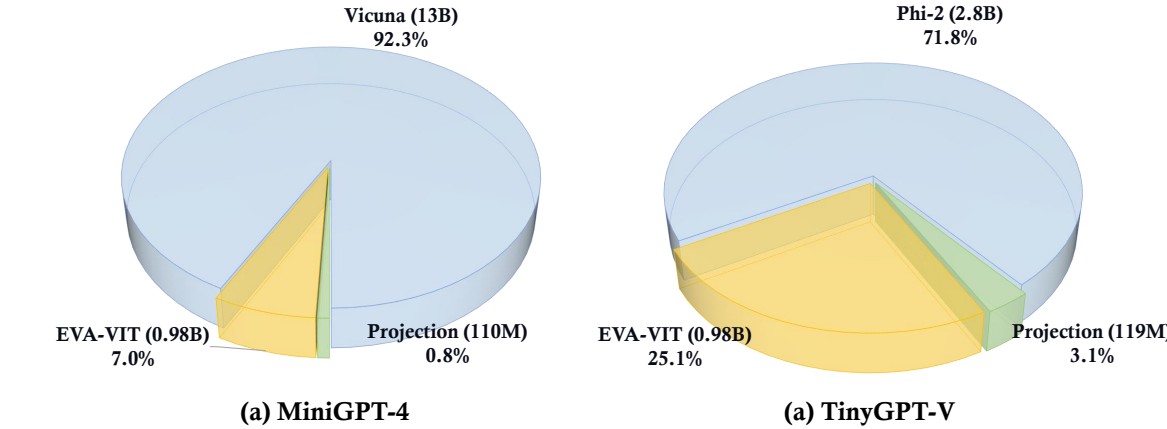

Figure 2: (a) is the occupancy ratio of each component in MiniGPT-4, and (b) is the occupancy ratio of each component in TinyGPT-V. We have considerably narrowed down the occupancy ratio of the Language model in MLLMs.

derlying performance of large language models, which are integral to MLLMs, is pivotal. Models such as LLaVA-v1.5-13B and MiniGPT-v2 depend on high-capacity backbones which are Vicuna-13b-v1.5 (Zheng et al., 2023) and LLaMA2-7B-Chat (Touvron et al., 2023), respectively, necessitating a substantial number of parameters to effectively tackle complex tasks including IC and VQA.

We introduce TinyGPT-V, a novel model designed for efficient training and inference, requiring only 24GB of GPU memory for training and as little as 8GB of GPU or CPU memory for inference. This model makes use of the advanced large language model Phi-2 (Javaheripi et al., 2023) and incorporates pre-trained vision modules from (Li et al., 2023a) and CLIP (Radford et al., 2021) as its vision encoder, coupled with a mapping module to facilitate the integration of visual information. During training, TinyGPT-V adopts a novel training methodology focused on small pre-trained backbones, unlike any other MLLMs, utilizing the unique mapping module between the visual encoder and the language model as well as novelty normalization methods, while keeping all other components frozen. For its training dataset, TinyGPT-V employs the multi-tasks datasets, including LAION (Schuhmann et al., 2021), Conceptual Captions (Changpinyo et al., 2021; Sharma et al., 2018), SBU (Ordonez et al., 2011), and others (Lin et al., 2015; Schwenk et al., 2022; Hudson & Manning, 2019; Kiela et al., 2020; Lu et al., 2021; Gurari et al., 2018; Mao et al., 2016; Kazemzadeh et al., 2014; Yu et al., 2016).

In our study, we found that TinyGPT-V exhibits similar traits with GPT-4, especially when doing some VQA and image inference. With only 2.8 billion parameters of its language model, TinyGPT-V employs a unique quantization process, like using 8-bit quantization, making it well-suited for local deployment and inference on 8GB mobile devices. This model represents a significant advancement in achieving a

balance between exceptional performance and efficiency in MLLMs, as shown in Figure 1. Our work not only aims to enable the community to develop more cost-effective, efficient, and high-performing MLLMs for widespread real-world applications but also introduces a training framework optimized for small pre-trained backbones.

## 2. Related Work

**Advanced language model.** The evolution of language models has been marked by significant milestones, starting with early successes like GPT2 (Radford et al., 2019) and BERT (Devlin et al., 2018) in natural language processing (NLP). These foundational models set the stage for the subsequent development of vastly larger language models, encompassing hundreds of billions of parameters. This dramatic increase in scale has led to the emergence of advanced capabilities as seen in models like GPT-3 (Brown et al., 2020), Chinchilla (Hoffmann et al., 2022), OPT (Zhang et al., 2022), and BLOOM (Workshop et al., 2022). These large language models (LLMs) have been instrumental in further advancements in the field. For instance, Chat-GPT (OpenAI, 2022) and InstructGPT (Ouyang et al., 2022) leverage these powerful models to answer diverse questions and perform complex tasks such as coding. The introduction of open-source LLMs like LLaMA (Touvron et al., 2023) has further propelled research in this area, inspiring subsequent developments like Alpaca (Taori et al., 2023), Vicuna (Chiang et al., 2023). These models fine-tune the LLaMA model with additional high-quality instruction datasets, showcasing the versatility and adaptability of LLM frameworks.Among the most notable recent advancements are Phi (Li et al., 2023b) and its successor, Phi-2 (Javaheripi et al., 2023). These models have demonstrated exceptional performance, rivaling or even surpassing models up to 25 times larger in scale. This indicates a significant shift in

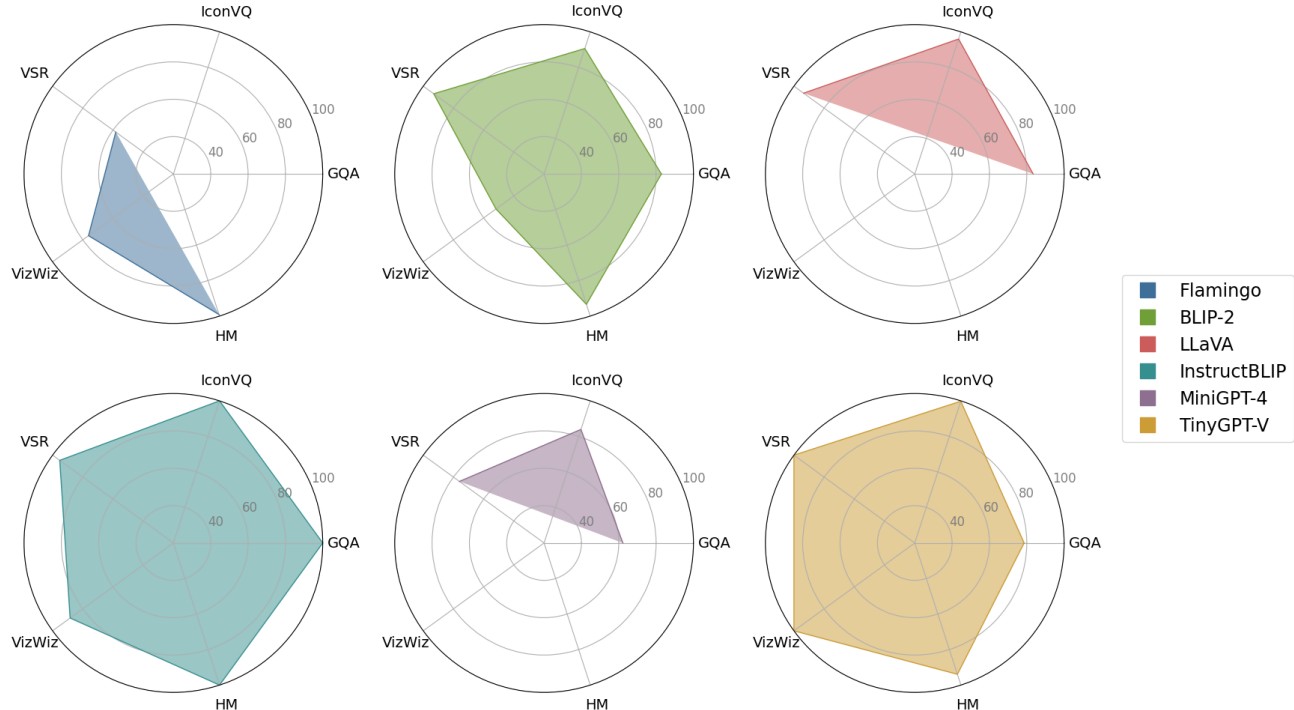

Figure 3: Compared to other general-purpose MLLMs, our TinyGPT-V achieves the same performance as 13B or 7B models in a variety of visual language tasks.

the landscape of language modeling, emphasizing efficiency and effectiveness without necessarily relying on sheer size.

**Multimodal language model.** In recent years, the trend of aligning visual input to large language models for vision-language tasks has gained significant attention (Chen et al., 2022; Tsimpoukelli et al., 2021; Alayrac et al., 2022; Li et al., 2023a; Liu et al., 2023b;a; Zhu et al., 2023; Chen et al., 2023). Seminal works like VisualGPT (Chen et al., 2022) and Frozen (Tsimpoukelli et al., 2021), which utilized pre-trained language models for image captioning and visual question answering. This approach was further advanced by models such as Flamingo (Alayrac et al., 2022), which incorporated gated cross-attention mechanisms to align pre-trained vision encoders and language models, training on vast image-text pairs. BLIP-2 (Li et al., 2023a) introduced an efficient Q-Former for aligning visual and language modalities. These groundbreaking studies have paved the way for further innovations in the field, leading to the development of models like LLaVA (Liu et al., 2023b) and MiniGPT4 (Zhu et al., 2023), and their subsequent iterations, LLaVA-v1.5 (Liu et al., 2023a), MiniGPT-v2 (Chen et al., 2023), ArtGPT-4 (Yuan et al., 2023), instruction GPT-4 (Wei et al., 2023) and Instruction Mining (Cao et al., 2023). These models have demonstrated advanced multimodal capabilities through instruction tuning, showcasing remarkable generalization abilities.

## 3. Method

We briefly introduce our vision-language model, TinyGPT-V, followed by an analysis of its structure, culminating in a detailed description of the training process for each stage.

### 3.1. Model Architecture

In this subsection, we present the architecture of TinyGPT-V, which consists of a visual encoder, projection layers, and a large language model, as shown in Figure 4.

**Visual encoder backbone.** In the TinyGPT-V, it utilizes EVA (Fang et al., 2022) of the ViT serves as the visual foundation model, which remains inactive during the entire training process. Our model operates at an image resolution of 224x224 for Stages 1, 2, and 3, and at 448x448 for Stage 4. The positional encoding is enhanced to accommodate the increased image resolution which is known as the Relative Position Bias (Dufter et al., 2021). It enhances the model's understanding of the spatial relationships between elements in an image.

**Projection layers.** The Projection layers embed visual features extracted by the visual encoder into the language model, enhancing the model's ability to process image-based information. We adopt the Q-Former layers from the BLIP-2 architecture (Li et al., 2023a) as the initial pro-

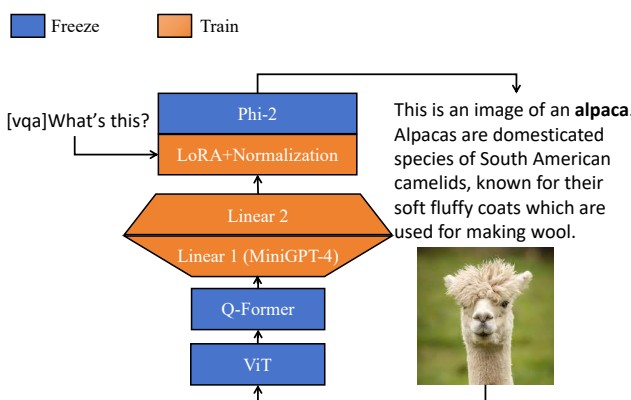

Figure 4: Architecture of TinyGPT-V. The model takes a visual backbone, which remains frozen during all training phases. We concatenate Q-Former layer visual output tokens from ViT backbone and project them into Phi-2 language model space via two linear projection layers.

jection layer, aiming to leverage the full potential of the pre-trained BLIP system within visual language models. This strategy significantly reduces the number of parameters needing training. The second and third layers are linear projection layers, designed to bridge the dimensionality gap between the Q-Former output and the language model's embedding layer, thereby aligning visual tokens more effectively with the language model's hidden space. As shown in Figure 6, to expedite TinyGPT-V's training, we initially use a pre-trained Linear Projection from MiniGPT-4 (Vicuna 7B) as the second layer. We then introduce an additional linear projection layer, initialized with a Gaussian distribution, as the third layer to seamlessly integrate into the hidden space of the Phi-2 model.

**Large lanuguage model backbone.** Our TinyGPT-V large language model is built upon the Phi-2 model (Javaheripi et al., 2023) as its backbone. Phi-2, a 2.7 billion-parameter language model, exhibits exceptional reasoning and language comprehension abilities, achieving state-of-the-art performance among language models with fewer than 13 billion parameters. In complex benchmarks, Phi-2 either matches or exceeds the performance of models up to 25 times its size. We primarily use Phi-2's linguistic abilities to do various vision-language tasks. Specifically, for vision reasoning tasks that involve spatial location identification, we instruct the linguistic model to generate textual descriptions of what will happen in the next scenario, representing their objects' coordinates, as shown in Table 8.

**Normalization and LoRA for TinyGPT-V.** In Section 4.4, we conclude that training smaller-scale large language models for transfer learning, especially across different modalities (e.g., from text to image), poses significant challenges.

Our studies indicate that these smaller models are prone to encountering NaN or INF values during multimodal data computations. This issue often leads to a computational loss value of NaN, thereby causing failure in the initial batch forward propagation. Moreover, the limited number of trainable parameters in these models may lead to gradient vanishing during training. To mitigate these problems, as depicted in Figure 5 (c), we incorporate the post-norm and input norm mechanisms from LLaMA-2, applying RMS Norm after each Multi-Head Attention Layer (MHA) to normalize data for downstream processing. In addition, we have to update all layer norms in the Phi-2 model to improve training stability, as detailed in the subsequent equation.

$$\text{LayerNorm}_{input}(x_{hidden}) = \gamma \frac{x_{hidden} - \mu}{\sqrt{\sigma^2 + \epsilon}} + \beta \quad (1)$$

Where, $x_{hidden}$ is the input of this layer, $\mu$ and $\sigma^2$ are the mean and variance of the inputs to the layer, respectively, $\epsilon$ is a small number to prevent division by zero, $\gamma$ and $\beta$ are trainable parameters.

$$\text{RMSNorm}(x_{post}) = \frac{x_{post}}{\sqrt{\frac{1}{N} \sum_{i=1}^{N} x_i^2 + \epsilon}} \quad (2)$$

where $x_{post}$ is the input after MHA, $N$ is the dimension of $x_{post}$.

Furthermore, (Henry et al., 2020) have underscored the vital role of Query-Key Normalization in low-resource learning scenarios. Hence, as show in Figure 5 (d), we have incorporated Query-Key Normalization into the Phi-2 model, as detailed in the following equation.

$$\text{Attention}(Q, K, V) = \text{softmax}\left(\frac{\text{LayerNorm}(Q)\text{LayerNorm}(K)^T}{\sqrt{d_k}}\right) V \quad (3)$$

where $d_k$ denotes the dimension of $Q$ or $K$.

The structure of the LoRA mechanism (Hu et al., 2021) is show in Figure 5 (a), which is an efficient fine-tuning method in parallel to the frozen pre-training weights as shown in Figure 5 (c), which does not increase the inference time consuming for large language models and is easier to optimize.

### 3.2. Training Stages

In this subsection, the four-stage training process of TinyGPT-V will be described.

**Warm-up training for the first training stage.** During the initial pretraining stage, TinyGPT-V is taught vision-language understanding using large datasets of aligned

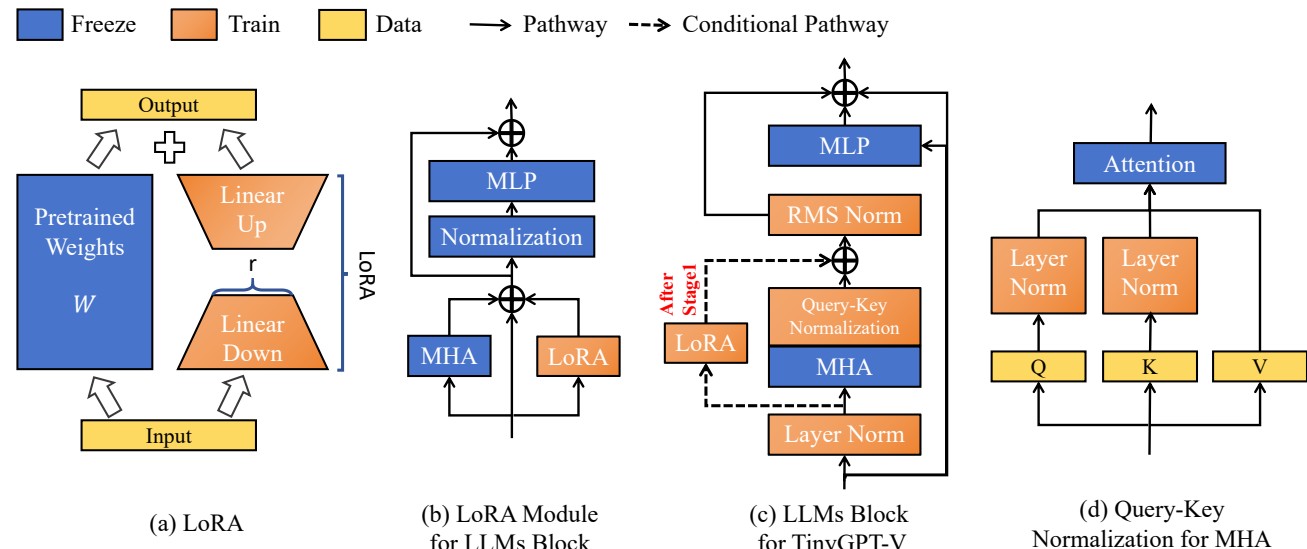

Figure 5: (a) represents the structure of LoRA, (b) represents how LoRA can efficiently fine-tune large language models (LLMs) in natural language processing, (c) represents the structure of LLMs for TinyGPT-V, and (d) represents the structure of QK Normalization.

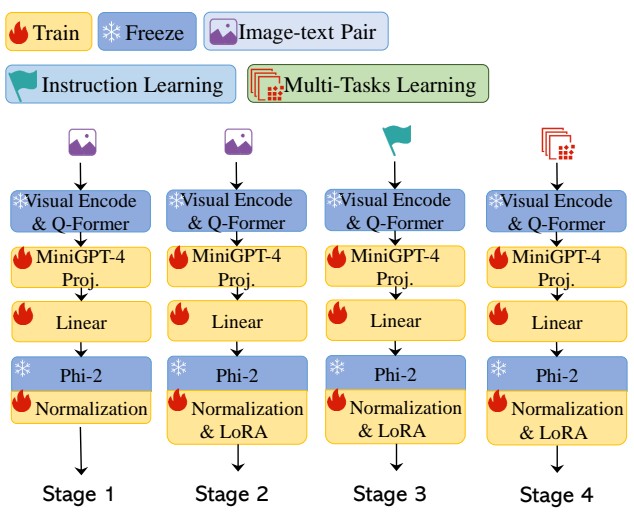

Figure 6: The training process of TinyGPT-V, the first stage is warm-up training, the second stage is pre-training, the third stage is instruction fine-tuning, and the fourth stage is multi-task learning.

image-text pairs. The model identifies the output from the projection layers as a soft prompt directing it to create relevant texts and to allow large language models to accept inputs from the image modality. The pretraining process uses a dataset combination of Conceptual Caption, SBU, and LAION, involving 20000 training steps covering about 5 million image-text pairs.

**Pre-training for the second training stage.** Following the initial training stage, the large language model becomes equipped to process image modality inputs. To guarantee more consistent performance as the model transitions into the subsequent training stage, we re-employ the dataset from the first stage, specifically for training the LoRA module.

**Instruction tuning for the third training stage.** We fine-tuned this TinyGPT-V model using a selection of image-text pairings from MiniGPT4 or LLaVA, which included instructions like "*###Human: <ImageHere></Img> Take a look at this image and describe what you notice.###Assistant:.*". We used a uniform template inclusive of a randomly chosen prompt that improved the model's capacity for generating responses that were consistent and sounded more natural.

**Multi-task learning in the fourth training stage.** The fourth training stage of TinyGPT-V focuses on enhancing its conversation ability as a chatbot by tuning the model with more multi-modal instruction datasets as shown in Table 1, including LLaVA, Flickr30k, a mixing multi-task dataset, and Unnatural Instruction using multi-tasks template as detailed in appendix A. The LLaVA dataset is utilized for multi-modal instruction tuning with detailed descriptions and complex reasoning examples. The Flickr30k dataset is used to improve grounded image caption generation and object parsing and grounding capabilities. Additionally, a mixing multi-task dataset is created to improve the model's handling of multiple tasks during multi-round conversations. Finally, to recover the language generation ability, the Unnatural Instruction dataset is added to the third-stage training of TinyGPT-V.

| Data types | Dataset | Stage 1 | Stage 2 | Stage 3 | Stage 4 |
|---|---|:---:|:---:|:---:|:---:|
| Image-text pair | LAION, CC3M, SBU | ✓ | ✓ | ✗ | ✗ |
| Instruction tuning | MiniGPT-4 Stage2 for CC & SBU | ✗ | ✗ | ✓ | ✗ |
| Caption | Text Captions, COCO Captions | ✗ | ✗ | ✗ | ✓ |
| REC | RefCOCO, RefCOCO+, RefCOCOg, Visual Genome | ✗ | ✗ | ✗ | ✓ |
| VQA | GQA, VQAv2, OK-VQA, AOK-VQA, OCR-VQA | ✗ | ✗ | ✗ | ✓ |
| Multimodal instruction | LLaVA dataset, Flickr30k, Multi-task conversation | ✗ | ✗ | ✗ | ✓ |
| Langauge dataset | Unnatural Instructions | ✗ | ✗ | ✗ | ✓ |

Table 1: The full list of datasets used by TinyGPT-MoE during training.

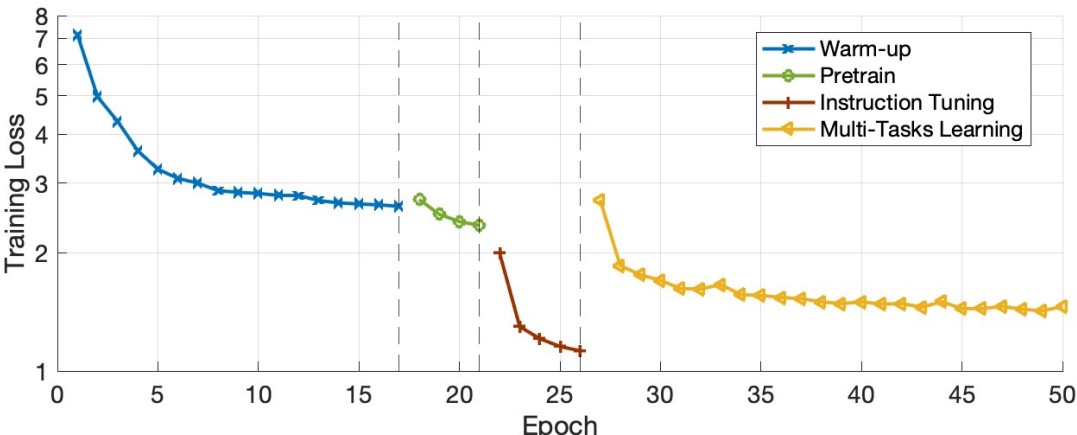

Figure 7: Changes in loss during the training stage of TinyGPT-V.

## 4. Experiments

In this section, we describe the training and evaluation methods in detail.

### 4.1. Training

**Experimental setting.** The experimental environment for this study was established with a single NVIDIA RTX 3090 GPU, equipped with a substantial 24GB of VRAM. The central processing was handled by an AMD EPYC 7552 48-core Processor, offering 15 virtual CPUs. Memory allocation was set at 80GB, ensuring sufficient capacity for handling large datasets. The software environment was standardized on PyTorch version 2.0.0, with CUDA 11.8 support, facilitating optimized tensor operations on the GPU.

**Training process.** In our experimental process, we meticulously orchestrated the training of our model through four distinct stages, each characterized by specific learning rate strategies and loss profiles, as shown in Figure 7.

Stage 1: Spanning 17 epochs, with each epoch consisting of 1000 iterations, we employed a dynamic learning rate approach. The learning rate commenced at 1e-5 at the beginning of each epoch and gradually ascended to 1e-4 by the epoch's end. This pattern was consistently applied across all

17 epochs. The training loss exhibited a steady decline, starting from 7.152 and progressively tapering down to 2.620, reflecting the model's increasing proficiency in learning from the data. The purpose of this stage is to be able to make the Phi-2 model in TinyGPT-V react in some way to the input of the imaging modality. The alignment of text and image in the semantic space is done.

Stage 2: Comprising 4 epochs, each with 5000 iterations, this stage introduced the "linear_warmup_cosine_lr" (He et al., 2018; Goyal et al., 2018) learning rate schedule. We initiated a warmup phase of 5000 steps, where the learning rate linearly increased from 1e-6 (warmup_lr) to 1e-4 (init_lr), followed by a cosine decay down to a minimum learning rate of 8e-5. This phase saw a consistent reduction in loss, starting at 2.726 and culminating at 2.343. The purpose of this stage is to enable the LoRA module to play a role in multimodal data, further reducing the model's loss on image-text pairs and improving the model's ability to learn from the data.

Stage 3: This stage lasted for 5 epochs, each with 200 iterations. We maintained the "linear_warmup_cosine_lr" schedule, with a warmup phase of 200 steps. The learning rate began at 1e-6, ascending to 3e-5 (init_lr), before decaying to 1e-5 (min_lr). The loss values reflected significant

improvements, starting at 1.992 and reducing to 1.125. The purpose of this stage is to allow TinyGPT-V to accept both verbal and image modal inputs and produce responses to them. After this stage of training TinyGPT-V has been able to perform most of the image answering tasks.

Stage 4: The final stage stretched over 50 epochs, each comprising 1000 iterations. We adhered to the "linear_warmup_cosine_lr" schedule with a 1000-step warmup phase. The learning rate was initiated at 1e-6, reaching up to 1e-5 (init_lr), and then experiencing a cosine decay to a minimum of 8e-5. The training loss values displayed a consistent downward trajectory, beginning at 2.720 and ultimately reaching as low as 1.399. The purpose of this stage is to allow TinyGPT-V to perform various tasks such as VQA or VSR tasks at the same time, increasing the generalization performance of TinyGPT-V on multimodal tasks.

## 4.2. Evaluation

**Evaluation datasets.** GQA (Hudson & Manning, 2019) is a dataset for real-world visual reasoning and compositional question answering, featuring a powerful question engine that generates 22 million diverse reasoning questions. VSR (Liu et al., 2023b) comprises over 10k natural text-image pairs in English, encompassing 66 types of spatial relations. IconQA (Lu et al., 2021) with 107,439 questions aimed at challenging visual understanding and reasoning in the context of icon images, encompassing three sub-tasks (multi-image-choice, multi-text-choice, and filling-in-the-blank). VizWiz (Gurari et al., 2018) is a collection of more than 31,000 visual queries, each derived from a photo taken by a visually impaired individual using a smartphone, accompanied by a vocalized question regarding the image, and supplemented with 10 answers sourced from a crowd for each query. The Hateful Memes dataset (HM) (Kiela et al., 2021), developed by Facebook AI, is a comprehensive multimodal collection specifically designed for the detection of hateful content in memes, combining both image and text elements, and comprises over 10,000 newly created multimodal examples.

**Visual question answering results.** As shown in Table 2, it becomes evident that TinyGPT-V, a language model with only 2.8 billion parameters, exhibits notably competitive performance across multiple benchmarks, closely rivaling models with nearly 13 billion parameters. Specifically, in the VSR (Visual-Spatial Reasoning) zero-shot task, TinyGPT-V outshines its counterparts by securing the highest score of 54.7%. This is particularly impressive considering its parameter size is approximately 4.6 times smaller than other leading models such as BLIP-2, LLaVA, and InstructBLIP. In the GQA benchmark, while TinyGPT-V scores are 38.9%, it lags behind the highest score achieved by InstructBLIP, which is 49.5%. However, TinyGPT-V shows robust perfor-

| TinyGPT-V and others answer example compare | |
|---|---|
| Ures | [vqa] where should I hide in this room when playing hide and seek |
| LLaVA-1.5 | hide behind the bookshelf |
| MiniGPT-v2 | behind couch |
| GPT-4V | Behind the Couch
Under the Table
Inside the Bookshelf
Behind the Curtains
Behind the TV |
| TinyGPT-V | under couch |

Figure 8: Comparison of reasoning answers from different Models. Text in red indicates incorrect suggestions. The TinyGPT-V's answer was short and precise.

mance in the IconVQ challenge, attaining a score of 44.7%, just 0.1% short of InstructBLIP's leading score of 44.8%. Similarly, in the VizWiz task, TinyGPT-V demonstrates commendable capabilities with a score of 37.8%, which, is not only the highest but is notable given its reduced parameter count. In the context of the Hateful Memes (HM) dataset, TinyGPT-V matches InstructBLIP's top score of 57.5% with its own score of 54.0%, again underscoring its efficiency and capacity to compete with models of larger scales. Overall, TinyGPT-V's performance across these diverse and challenging benchmarks is striking, especially when considering its parameter efficiency

## 4.3. Qualitative Evaluation

The comparative analysis revealed TinyGPT-V's distinct advantage in delivering concise and accurate visual interpretations. In the reasoning task to find a hiding spot during a game of hide and seek, TinyGPT-V demonstrated its superior capability by providing a singular, viable suggestion: 'under couch'. This contrasts with other models that either offered multiple options, some of which were incorrect as indicated by the text in red (e.g., GPT-4V suggesting 'Inside the Bookshelf'), or specified less practical hiding spots. When asked about potential activities in an image with an alligator, TinyGPT-V suggested a cautious response without speculating beyond what was visible. In contrast, other models, like LLaVA-1.5, provided extended narratives that introduced assumptions not directly inferred from the image. Similarly, in describing a soccer match scene, TinyGPT-V's response was succinct and focused on the key elements, avoiding the inaccuracies noted in MiniGPT-v2's account, which incorrectly identified multiple soccer balls on the

| Method | LLM Parameters | GQA | VSR (zero-shot) | IconVQ (zero-shot) | VizWiz (zero-shot) | HM (zero-shot) | Average |
|---|---|---|---|---|---|---|---|
| Flamingo | 9B | - | 31.8 | - | 28.8 | 57.0 | 39.20 |
| IDEFICS  (Laurençon et al., 2023) | 7B | - | 38.4 | - | 35.5 | - | 37.05 |
|  | 65B | - | 45.2 | - | 36.0 | - | 39.60 |
| BLIP-2 | 13B | 41.0 | 50.9 | 40.6 | 19.6 | 53.7 | 41.16 |
| LLaVA | 13B | 41.3 | 51.2 | 43.0 | - | - | 45.17 |
| InstructBLIP | 13B | 49.5 | 52.1 | **44.8** | 33.4 | **57.5** | **47.45** |
| MiniGPT-4 | 13B | - | - | 35.9 | - | - | 35.90 |
| BLIVA (Hu et al., 2023) | 7B | - | - | 44.8 | 31.4 | 55.6 | 41.15 |
| LLaVA-Phi (Zhu et al., 2024) | 2.8B | - | - | 54.1 | 37.6 | - | 43.15 |
| MoE-LLaVA (Lin et al., 2024)* | 1.8B×4 | **61.5** | - | - | 32.6 | - | **47.50** |
| **Ours** | | | | | | | |
| TinyGPT-V (Phi-2) | 2.8B | 38.9 | **54.7** | 44.7 | **37.8** | 54.0 | **46.02** |
| TinyGPT-V (Phi-1.5) | 1.3B | 34.3 | 35.8 | 37.2 | 28.4 | 50.3 | 37.2 |

Table 2: Comparative performance of TinyGPT-V and other MLLMs across multiple visual question answering benchmarks. *It is worth noting that MoE-LLaVA is required 8xA100-80G for training.

| Method | TinyGPT-V | LLaVA | MiniGPT-4 |
|---|---|---|---|
| seconds per words | 0.067 | 0.426 | 0.300 |
| inference occupancy (8-bit) | 5.6GB | 22GB | 23.5GB |

Table 3: Comparison of inference time and inference occupancy about devices.

| Method | Stage 1 Loss | Stage 2 Loss | Stage 3 Loss | Stage 4 Loss |
|---|---|---|---|---|
| TinyGPT-V | 2.620 | 2.343 | 1.125 | 1.330 |
| w/o LoRA | 2.620 | - | Gradient Vanish | - |
| w/o Input Layer Norm | 2.839 | 2.555 | 1.344 | Gradient Vanish |
| w/o RMS Norm | 2.747 | Gradient Vanish | - | - |
| w/o QK Norm | Gradient Vanish | - | - | - |

Table 4: Importance of each module in TinyGPT-V at each stage of training.

pitch. These examples, as tabulated in Table 5 and Table 7, illustrate TinyGPT-V's superior performance in generating brief yet precise responses, underscoring its practicality for rapid and reliable visual question answering. For efficient evaluation, as shown in table 3, TinyGPT-V operates at the fastest pace, taking only 0.067 seconds to generate a word, which suggests upper efficiency in processing speed compared to LLaVA and MiniGPT-4. On the other hand, LLaVA exhibits a significantly slower word generation time at 0.426 seconds per word, coupled with a higher memory occupancy of 22GB. MiniGPT-4, with a generation time of 0.300 seconds per word and a memory usage of 23.5GB.

### 4.4. Ablation Study

As shown in Table 4, the full TinyGPT-V model achieves low loss across all stages, but the removal of key modules leads to significant training issues. Without the LoRA module, there's a gradient vanish starting from Stage 3. Omitting Input Layer Norm increases loss notably (to 2.839 in Stage 1) and causes gradient vanishing in Stage 4. Without RMS Norm, the model sees an elevated loss in Stage 1 (2.747) and faces early gradient vanishing in Stage 2. The absence of QK Norm results in immediate gradient vanish. This data clearly illustrates each module's crucial role in preventing gradient vanishing and maintaining low loss throughout the training process.

Furthermore, our reveal a notable trend: the smaller the large language model used for transfer learning (particularly in transitioning from text-to-image modality), the more challenging the training process becomes. We observed a pronounced need for additional normalization layers to stabilize the training, especially when scaling down from larger models like Vicuna-13B to smaller ones like Phi-2 (2.7B), Phi-1.5 (1.3B), and other small backbones as detailed in the Appendix B.

## 5. Conclusion

In this study, we introduce TinyGPT-V, a parameter-efficient MLLMs tailored for a range of real-world vision-language applications. Our model innovatively builds on the compact yet powerful Phi-2 small language model framework. This approach results in TinyGPT-V delivering exceptional outcomes in diverse benchmarks like visual question-answering and referring expression comprehension while keeping computational demands manageable. Remarkably, TinyGPT-V can be trained on a 24G GPU and deployed on an 8G device, demonstrating a significant advancement in creating cost-effective, efficient, and potent MLLMs. This paper marks a contribution towards crafting smaller, yet robust multimodal language models for practical, real-world use cases. We envision that our work will catalyze further explorations into developing compact MLLMs for diverse applications.

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

## A. Multi-task Instruction Template

To reduce ambiguity in training a unified multimodal model for diverse tasks such as visual question answering, image captioning, referring expression comprehension, generation, and object parsing and grounding, we employed task-specific tokens from MiniGPT-v2 within a multitask instruction template. This template, derived from the LLaMA-2 conversation template (Touvron et al., 2023), includes a general input format comprising image features, a task identifier token, and an instruction input. We incorporated six distinct task identifiers, each associated with a particular task. For tasks necessitating the identification of spatial locations of referred objects, the model uses textual representations of bounding boxes with coordinates normalized between 0 and 100. Overall, MiniGPT-v2's unique task-specific tokens enhance task disambiguation, leading to more precise and effective task execution.

| TinyGPT-V and others answer example compare | |
|---|---|
| **Ures** | What might happen in this image in the next second
 |
| **LLaVA-1.5** | In the image, a man is sitting on a log next to a large alligator, and he is holding a camera. It is likely that the man is about to take a photograph or record a video of the alligator. The next second could involve the man capturing the alligator's behavior, appearance, or any other interesting moment. The alligator might also react to the man's presence, such as moving or showing curiosity, which could make the captured image or video more engaging and memorable. |
| **MiniGPT-v2** | In the next second, the man might be startled or surprised by the close-up shot of the crocodile in the water. This could cause him to jump back or react with caution, ensuring his safety in the presence of the wild animal. The image captures a moment of wonder and curiosity, highlighting the beauty of the natural environment while also acknowledging the potential danger that comes with being so close to a wild animal. |
| **GPT-4V** | In the next second, the man might quickly move away to maintain a safe distance from the alligator. Alligators can be unpredictable, and it's important to keep a safe distance from them at all times to avoid potential harm. The man seems to be taking a selfie with the alligator, which suggests that he is close to the animal. This is generally not advisable as it can be dangerous. It's always important to respect wildlife and observe it from a distance that is safe for both the person and the animal. |
| **TinyGPT-V** | The image shows a man holding a selfie camera next to a crocodile. This could be an unusual and unexpected sight, as crocodiles are not typically associated with human activities like taking selfies. lt's important for the man to stay safe and avoid any potential risks posed by the crocodile. |

Table 5: Comparison of prediction answers from different models. Text in red indicates incorrect suggestions. The TinyGPT-V's answer was short and precise.

## B. Small Backbones for Transfer Learning

As shown in Table 6, a striking pattern emerges from the data: smaller LLMs exhibit heightened sensitivity to the removal of these modules, with a pronounced tendency towards training difficulties, such as gradient vanishing. For instance, the absence of LoRA in both Phi-1.5 and TinyLLaMA resulted in an immediate cessation of training progress post-Stage 1, indicating a critical reliance on this module for sustaining training in smaller models. Similarly, the exclusion of QK Norm led to gradient vanishing at the earliest stage across all smaller LLMs, underscoring its essential role in the initial phases

of training. Moreover, the sequential progression in training losses across stages for models without these modifications demonstrates a clear degradation in training efficiency and effectiveness. For example, the removal of Input Layer Norm and RMS Norm not only heightened Stage 1 loss across Phi-1.5 and TinyLLaMA but also precipitated gradient vanishing in later stages, showcasing the compound impact of these modules on model stability and learning capability. This analysis incontrovertibly highlights a fundamental challenge in training smaller LLMs for migration to MLLMs: the absence of key architectural and normalization modules severely impedes their training process, making them more prone to early training halts and efficiency losses. The results underscore the necessity of these components in supporting the stability and gradual learning progression of smaller LLMs, thus illuminating a pivotal consideration for developers aiming to optimize the training framework for seamless model.

| LLM | Stage 1 Loss | Stage 2 Loss | Stage 3 Loss | Stage 4 Loss |
|---|---|---|---|---|
| Phi-2 (2.7B) | 2.620 | 2.343 | 1.125 | 1.330 |
| Phi-1.5 (1.3B) | 3.420 | 3.043 | 1.525 | 1.730 |
| w/o LoRA | 3.420 | - | Gradient Vanish | - |
| w/o Input Layer Norm | 3.555 | 3.221 | 1.544 | Gradient Vanish |
| w/o RMS Norm | 3.557 | Gradient Vanish | - | - |
| w/o QK Norm | Gradient Vanish | - | - | - |
| TinyLLaMA (1.1B) | 3.529 | 3.053 | 1.371 | 1.830 |
| w/o LoRA | 3.529 | - | Gradient Vanish | - |
| w/o Input Layer Norm | 3.611 | 3.331 | 1.444 | Gradient Vanish |
| w/o RMS Norm | 3.607 | Gradient Vanish | - | - |
| w/o QK Norm | Gradient Vanish | - | - | - |

Table 6: Importance of each module and different LLMs in TinyGPT-V training framework at each stage of training.

| TinyGPT-V and others answer example compare | |
|---|---|
| **Ures** | What's in the image?  |
| **LLaVA-1.5** | The image shows a group of men playing soccer on a field. They are actively engaged in the game, with some players running towards the soccer ball. |
| **MiniGPT-v2** | The image features a lively and competitive soccer game with several players on the field. Some of the players are wearing jerseys, while others have their own unique outfits. There are at least eight players in total, with three of them being the main focus of the scene.
One player is kicking a soccer ball, with another player nearby trying to block the shot. A third player is also present, playing offense. In addition, there are two soccer balls visible on the field, one closer to the left side and the other near the center of the scene.
The players are engaged in intense action, showcasing their skills and determination to score. The atmosphere appears to be a high-energy match with a mix of excitement and competition among the players. |
| **GPT-4V** | The image shows a moment from a soccer match. In the foreground, there's a player in a blue uniform who appears to be dribbling the ball with close control, evading two opponents in white uniforms. The player in blue is crouched low to the ground, demonstrating agility and focus, which suggests he is in the midst of making a strategic play, possibly attempting to bypass the defenders. In the background, other players can be observed, and the stands are filled with spectators, indicating that this is a professional match likely taking place in a large stadium. The attire and hairstyles suggest the photo may be from a past era, possibly the 1980s or 1990s. |
| **TinyGPT-V** | The image shows a soccer game with several people playing, including at least one person in a blue shirt. The players are engaged in a friendly match, showcasing their skills and teamwork. |

Table 7: Comparison of descriptive answers from different Models. Text in red indicates incorrect suggestions. The TinyGPT-V's answer was short and precise.