# OpenReview forum: "TinyGPT-V: Efficient Multimodal Large Language Model via Small Backbones"
_ICML.cc/2024/Workshop/WANT — WANT@ICML 2024 Poster_

### Official Review · Reviewer_Nbce · 2024-06-13
**Replacing LLMs with smaller ones with increased training**

**Confidence:** 4

**Summary:**

This paper explains a method to replace the large Vicuna-13B model with Phi-2 (2.8B) model. The authors propose to do this by adding a few previously-known components to the MLLM and training the model for longer.

**Strengths:**

1. Clear objective and methodology: The authors have a single objective - to reduce the size of LLM in the overall model. The authors focus their attention around this itself and do the needful (training for longer and using pretrained components) for the same.
2. Usage of pretrained components from different models: The authors employ different pretrained components from BLIP-2, which is a much larger pretrained model. Intuitively, this helps to transfer the knowledge from the larger models trained on larger corpora to the smaller setup.
3. Clear description of additional components and training setup: The authors describe and analyze the training setup sufficiently.

**Weaknesses:**

1. Despite the success of this proposed framework, the added components are elaborate and complicated. Ablations with simple components, like just a single MLP, would have justified the usage of these additional components more.
2. Large amounts of training data and substantially longer training routines were used in this work. This is not directly a weakness and is partially justified due to the usage of a smaller LLM, however, a more stringent usage of data would have made this a more favorable framework for researchers.

**Limitations:**

1. Large amounts of data shadow the effectiveness of the smaller LLM.
2. Diagrams could have been better, since they seem tacky and can be improved.

**Suggestions:**

1. A strong analysis of the individual effectiveness of each training dataset could significantly improve the confidence in the paper.
2. The authors could optionally explore distillation-like methods.
3. The authors could explore a simpler training setup.

The current draft is a good one, and can be accepted in its current condition. The above are just suggestions to make the draft better.

---

### Official Review · Reviewer_1ky8 · 2024-06-13
**Review of TinyGPT-V paper**

**Confidence:** 5

**Summary:**

Paper introduces TinyGPT-V, a new, open-source model designed to be efficient and effective for vision-language tasks. TinyGPT-V demonstrates that smaller, optimized models can be both high-performing and resource-efficient, making advanced AI more accessible. This study highlights the potential of using smaller model architectures in developing efficient multimodal large language models.

**Strengths:**

All sections are well-structured with enough information. The abstract provides a concise summary of the paper's objectives, methods, results, and significance, which is essential for scientific papers. The methodology section includes sufficient technical details, making it reproducible for other researchers, which is a crucial aspect of scientific research. The inclusion of figures and tables enhances the clarity of the results and supports the claims made in the paper. The conclusion summarizes the findings, emphasizes the significance of the work, and suggests potential future directions.

**Weaknesses:**

1) The paper claims that TinyGPT-V achieves comparable results to larger models but does not provide a detailed comparative analysis with specific benchmarks or metrics for all competing models. It would be beneficial to see more thorough comparisons with the most relevant existing models, including any edge cases where TinyGPT-V might not perform as well.
2) The paper mentions various tasks (e.g., image captioning, visual question answering) but does not provide a deep dive into the model's performance across a wide range of tasks. There should be more evidence on how well TinyGPT-V generalizes across different types of vision-language tasks.

---

### Official Review · Reviewer_kHHA · 2024-06-14
**TinyGPT-V: Efficient Multimodal Large Language Model via Small Backbones**

**Confidence:** 3

**Summary:**

This paper presents TinyGPT-V, an open source parameter-efficient MLLM multimodal language model designed for vision-language tasks. Built on the Phi-2 small language model framework, TinyGPT-V is better in lots of benchmarks, such as visual question-answering and referring expression comprehension, while maintaining low computational needs and being open source. It can be trained on a 24gb GPU and deployed on an 8gb device. They also explained in detail their training technology, which uses a four-stage training process, including a language model and a visual encoder.

**Strengths:**

* The paper is easy to understand and follow
* They explained the method so that people would be able to replicate it.
* Method is an open source.
* By using ViT and Phi2 models with frozen weights and training only the component arrangements (linear layers and LoRA parameters) the system becomes highly flexible for future applications.

**Weaknesses:**

* The model's smaller scale, while it is great for computational efficiency, may limit its ability to handle highly complex or nuanced tasks that larger models could manage potentially using the same training procedure as this paper suggests.

**Limitations:**

* TinyGPT-V's testing focuses on specific benchmarks, which may not fully represent its performance in diverse real-world scenarios.

**Suggestions:**

It would be good to test not only Phi-2 but also larger and more powerful language models. While Phi-2 performs well in tests, it might not do as well in real-life situations. Comparing it with bigger models could give us a better idea of how they work in everyday conversations.

---

### Meta-Review · Area_Chair_yr6z · 2024-06-18

**Recommendation:** Accept (Oral)
**Confidence:** 5

**Metareview:**

**Strengths**
- TinyGPT-V reduces the size and costs of MLLMs thus increasing the availability of this important technology.
- The paper is well written, and methods well described fostering reproducibility.
- TinyGPT-V is open sourced and thus useful for the broad community.

**Weaknesses**
- It would be useful to discuss TinyGPT-V limitations, where it is inferior to larger MLLMs.
- Evaluation should include a wider range of benchmarks and tasks.

**Summary**
This is a timely and important work for the community.

---

### Decision · Program_Chairs · 2024-06-18

**Decision:**

Accept (Poster)

**Comment:**

We thank the authors for their time and contribution to WANT and we are pleased to share that after the reviewing process the paper has been accepted. Congratulations! We encourage the authors to consider reviewers' feedback for the improvement of the camera-ready version. We hope to see you in person at the workshop and brainstorm on efficient training research together!